# A Distance Covariance-based Kernel for Nonlinear Causal Clustering in Heterogeneous Populations

**Alex Markham**                                                    ALEX.MARKHAM@CAUSAL.DEV
*Research Group Neuroinformatics, Faculty of Computer Science, University of Vienna*
*Mathematics of Data and AI, Department of Mathematics, KTH Royal Institute of Technology*

**Richeek Das**
*Department of Computer Science and Engineering, Indian Institute of Technology Bombay*

**Moritz Grosse-Wentrup**
*Research Group Neuroinformatics, Faculty of Computer Science, University of Vienna*
*Research Platform Data Science @ Uni Vienna*
*Vienna Cognitive Science Hub*

**Editors:** Bernhard Schölkopf, Caroline Uhler and Kun Zhang

## Abstract

We consider the problem of causal structure learning in the setting of heterogeneous populations, i.e., populations in which a single causal structure does not adequately represent all population members, as is common in biological and social sciences. To this end, we introduce a distance covariance-based kernel designed specifically to measure the similarity between the underlying nonlinear causal structures of different samples. Indeed, we prove that the corresponding feature map is a statistically consistent estimator of nonlinear independence structure, rendering the kernel itself a statistical test for the hypothesis that sets of samples come from different generating causal structures. Even stronger, we prove that the kernel space is isometric to the space of causal ancestral graphs, so that distance between samples in the kernel space is guaranteed to correspond to distance between their generating causal structures. This kernel thus enables us to perform clustering to identify the homogeneous subpopulations, for which we can then learn causal structures using existing methods. Though we focus on the theoretical aspects of the kernel, we also evaluate its performance on synthetic data and demonstrate its use on a real gene expression data set.

**Keywords:** graphical causal models; distance covariance; whole-graph embeddings; clustering.

## 1. Introduction

Learning causal relationships from observational and experimental data is one of the fundamental goals of scientific research, and causal inference methods are thus used in a wide variety of fields. The resulting variety of applications nevertheless share some common difficulties, such as causal inference from complex time-series data (Eichler, 2012) or the underlying causal structure being obscured by unmeasured confounders (Greenland et al., 1999). Another common difficulty, especially for applications in the biological and social sciences, is causal inference from heterogeneous populations (Xie, 2013; Brand and Thomas, 2013)—addressing this difficulty is our main motivation.

In general terms, we understand a heterogeneous population to be one whose members are not adequately described by a single model but rather better described by a collection of models. Within our context of causal structure learning, this means a population is heterogeneous if some samples are generated by different causal structures—we call this *structural* heterogeneity. We note that there

are other kinds of heterogeneity, such as that in samples generated by different joint distributions over the same causal structure, but these are outside our scope in this work.

A specific example of structural heterogeneity can be found in genetics: causal methods are used to learn the structure of gene regulatory networks (Altay and Emmert-Streib, 2010), and gene expression data from a single recording or experiment may include thousands of genes, many of which are involved in entirely different networks (Liu, 2015); thus, attempting to learn a single causal structure for all of the genes will obscure the fact that different sets of them have different structures.

The bulk of our work in this paper, and our main contribution, is to introduce the *dependence contribution kernel*, which facilitates a flexible and easily extensible approach to causal clustering: first perform clustering to identify structurally homogeneous subsets of samples, and then proceed with the actual learning task on each cluster. We prove that our kernel space is isometric to the space of causal ancestral graphs, and hence our kernel can be used to find clusters that minimize structural heterogeneity for causal structure learning tasks. Furthermore, the kernel is derived from the distance covariance (Székely et al., 2007), imbuing it with the ability to detect nonlinear dependence. It can easily be used in a wide array of clustering algorithms, such as $k$-means, DBSCAN, spectral clustering, or any other method that analogously makes use of a similarity (or distance) measure between samples (Filippone et al., 2008).

The rest of the paper is organized as follows: We finish this section by discussing some of the most relevant related work from the causal inference and statistics literature. All of Section 2 is devoted to the theory underlying our dependence contribution kernel, including a comparison of the familiar product-moment covariance with the distance covariance (Section 2.1), defining an equivalence class of causal models with a convenient representation in the kernel space (Section 2.2), and the actual definition of our kernel and proofs of its relevant properties (Section 2.3). Next, in Section 3, we supplement the preceding theoretical analysis of our kernel by evaluating its use in a combination of clustering, dimensionality reduction, and structure learning tasks on synthetic and real data. Finally, we conclude in Section 4 mentioning possible future work.

## 1.1. Related Work

Causal inference in heterogeneous populations sometimes refers to data-fusion (Bareinboim and Pearl, 2016), i.e., combining known homogeneous subpopulations and performing causal inference on the resulting heterogeneous population, or similarly, it can refer to meta-learning using known subpopulations (Sharma et al., 2019). Other times, it refers to estimating heterogeneous treatment effects (Xie et al., 2012; Athey and Imbens, 2015). However, in our case, the subpopulations are not known and we rather consider the problem of learning which samples come from which subpopulation, and these are differentiated according to structure instead of treatment effect.

Previous work on causal clustering has focused more on the causal modeling aspect, using stronger assumptions about the underlying structures to learn more detailed models. For example, Kummerfeld et al. (2014); Kummerfeld and Ramsey (2016) focus on causal clustering in measurement models, with the goal of clustering different features together to study their latent causal structure, based on tetrad constraints within the linear product-moment covariance matrix. Huang and Zhang (2019) define a class of causal models facilitating mechanism-based clustering, learning causal models both for clusters of samples as well as a shared one for all samples, assuming the underlying structures are linear non-Gaussian. Saeed et al. (2020) characterize distributions arising from mixtures of directed acyclic graph (DAG) causal models (i.e., causal models without latent or

selection variables), trying to learn both the component DAGs and a representation of how they are mixed. All of these approaches, like most causal inference methods, make specific (and for some applications, restrictive) assumptions about the underlying distributions or causal structures.

In contrast, our method is not tied to specific distributional assumptions such as linearity or (non)Gaussianity—we assume there are enough samples for statistical inference, as well as the usual causal Markov and faithfulness assumptions. For the first step, we cluster samples together if they (implicitly, in the kernel space) have similar nonlinear independence structures. For the second step, causal structure learning, any existing method (along with its corresponding assumptions) can in principle be used. In our gene expression data application (Section 3.2), the measurement dependence inducing latent (MeDIL) causal model framework (Markham and Grosse-Wentrup, 2020), which assumes the data consists of measurement variables that are causally connected only through latent variables, seems appropriate, however other applications can easily use other methods. For example, component and mixture DAGs (Saeed et al., 2020) can be better learned when one first knows which samples come from which component—clustering with our kernel ensures samples in different clusters come from different DAGs, and so using their method instead of the MeDIL framework would be a natural choice for applications in which a DAG (without any latents) is more appropriate.

Finally, there is some work from the statistics literature that sounds superficially similar to our distance covariance-based kernel but is conceptually quite different. Namely, another well-known measure of nonlinear independence, the Hilbert-Schmidt Independence Criterion (Gretton et al., 2005, 2007), is part of a class of reproducing kernel Hilbert space- (RKHS-) based dependence measures that Sejdinovic et al. (2013) show is equivalent to distance-based measures such as the distance covariance. Our dependence contribution kernel, unlike these, is not a dependence measure between features—it rather uses the distance covariance to measure the similarity of samples based on patterns in the dependence structure of their features, and is rather more like a whole-graph embedding (Cai et al., 2018; Maddalena et al., 2020).

## 2. Theory

### 2.1. Product-moment Covariance, Distance Covariance, and Dependence Contribution

Though there is more to causal relationships than probabilistic dependence, causal inference methods based on graphical models ultimately rely on at least implicitly learning conditional independence (CI) relations. CI relations can be estimated in many ways, with different dependence measures and tests each having their own theoretical guarantees and being better suited for distributions of various different kinds of data (e.g., categorical, discrete, or continuous) and with various kinds of relationships (e.g., linear, monotonic nonlinear, arbitrary nonlinear) and with different testing assumptions (see Tjøstheim et al., 2018, for a comprehensive overview).

A widely used measure of dependence is the *product-moment covariance*, often just called covariance, which is defined for two zero-mean random variables $X_1$ and $X_2$ as the scalar value $\mathrm{cov}(X_1, X_2) = \mathrm{E}[X_1 X_2]$. This can be extended from a pair of random variables to every pair of variables in a random vector, thus returning a matrix instead of a scalar. The covariance matrix for a vector of zero-mean random variables $\mathbf{X} = (X_1, \ldots, X_m)$ can be estimated from a set $S \in \mathbb{R}^{n,m}$ of $n$ samples as $\hat{\Sigma}_{\mathbf{X}} = \frac{1}{n} S^\top S$, and the $j, j'$-th value of $\hat{\Sigma}_{\mathbf{X}}$ is thus the estimate $\hat{\mathrm{cov}}(X_j, X_j')$.

Probabilistic independence (denoted $\perp\!\!\!\perp$) of two random variables implies that their product-moment covariance is zero, i.e., $X_j \perp\!\!\!\perp X_{j'} \implies \mathrm{cov}(X_j, X_{j'}) = 0$ (importantly, the inverse of this does not hold). Thus, the estimated product-moment covariance can be used in statistical hypothesis

testing for independence (Wasserman, 2013, Ch. 10): $X_j$ and $X_{j'}$ are taken to be independent if and only if $\hat{\text{cov}}(X_j, X_{j'})$ is sufficiently close to 0. However, this method has an important flaw: the product-moment covariance as a test statistic is only valid against *linear* dependence.

Székely et al. (2007) introduce the *distance covariance* to remedy this problem: random variables are probabilistically independent if and only if their distance covariance is zero, i.e., $X_j \perp\!\!\!\perp X_{j'} \iff \text{dCov}(X_j, X_{j'}) = 0$, resulting in the estimated distance covariance being a valid test statistic against all types of dependence. The distance covariance is related to the product-moment covariance by $\text{dCov}^2(X_j, X_{j'}) = \text{cov}(|X_j - X_j'|, |X_{j'} - X_{j'}'|) - 2\text{cov}(|X_j - X_j'|, |X_{j'} - X_{j'}''|)$, where $(X_j', X_{j'}')$ and $(X_j'', X_{j'}'')$ are independent and identically distributed (iid) copies of $(X_j, X_{j'})$ (Székely and Rizzo, 2014). The key intuition here is that the distances (e.g., $|X_j - X_j'|$) constitute a nonlinear projection, so that using the linear product-moment covariance in this projected space allows for the detection of nonlinear dependence in the original space.

Note that dCov is typically defined to be a scalar value when taken between two arbitrary-dimensional random vectors, but our restricted presentation of it above in terms of random variables is to make it more obviously analogous to the product-moment covariance between random variables. Thus, corresponding to $\hat{\Sigma}_{\mathbf{X}}$ for random vectors, we define the following:

**Definition 1** *Let $S \in \mathbb{R}^{n,m}$ be a set of $n$ samples from the vector of random variables $\mathbf{X} = (X_1, \ldots, X_m)$. For each $j \in \{1, \ldots, m\}$ and $i, i' \in \{1, \ldots, n\}$, define the pairwise distance matrix $D^j$, with values given by $D^j_{i,i'} := |S_{i,j} - S_{i',j}|$. Now define the corresponding doubly-centered matrices $C^j_{i,i'} := D^j_{i,i'} - \bar{D}^j_{i,\cdot} - \bar{D}^j_{\cdot,i'} + \bar{D}^j_{\cdot,\cdot}$, where putting a bar over the matrix and replacing an index $i$ or $i'$ with $\cdot$ denotes taking the mean over that index. Define the matrix $L \in \mathbb{R}^{n^2,m}$ so that each column is a flattened doubly-centered distance matrix, $L := (\text{vec}(C^1), \ldots, \text{vec}(C^m))$, where $\text{vec}(C^j)$ denotes "flattening" matrix $C^j$ into a column vector. Finally, the estimated distance covariance matrix over sample $S$ is defined as:* $\quad \hat{\Delta}_{\mathbf{X}} := \frac{1}{n^2} L^\top L$.

Analogous to $\hat{\Sigma}_{\mathbf{X}}$, the $j, j'$-th entry of $\hat{\Delta}_{\mathbf{X}}$ corresponds to $\hat{\text{dCov}}^2(X_j, X_{j'})$—indeed it is mathematically equivalent to computing each pairwise distance covariance value and then manually filling in the matrix. The novelty of our Definition 1 is in finding a matrix of pairwise values instead of a single value for the distance covariance between random vectors, which helps provide an intuition for our next definition:

**Definition 2** *Let $S \in \mathbb{R}^{n,m}$ be a set of $n$ samples from the vector of random variables $\mathbf{X} = (X_1, \ldots, X_m)$; note that we consistently use indices $i, i' \in \{1, \ldots, n\}$ and $j, j' \in \{1, \ldots, m\}$. Let $D \in \mathbb{R}^{n,n,m}$ denote the 3-dimensional array of stacked pairwise distance matrices defined by $D_{i,i',j} := |S_{i,j} - S_{i',j}|$, and use $C \in \mathbb{R}^{n,n,m}$ to denote these same distance matrices after being doubly-centered, i.e., $C_{i,i',j} := D_{i,i',j} - \bar{D}_{i,\cdot,j} - \bar{D}_{\cdot,i',j} + \bar{D}_{\cdot,\cdot,j}$, where replacing an index $i$ or $i'$ with $\cdot$ denotes the entire (lower-dimensional) subarray over that index, and writing a bar, $\bar{D}$, denotes taking the mean over that subarray. Then standardize the doubly-centered distances to get $Z_{i,i',j} := \frac{C_{i,i',j}}{\bar{D}_{\cdot,\cdot,j}}$. Finally, the* dependence contribution map, $\varphi : \mathbb{R}^m \to \mathbb{R}^{m,m}$*, is defined as*

$$\varphi(S_{i,\cdot}) := Z^\top_{i,\cdot,\cdot} Z_{i,\cdot,\cdot} - \mathcal{T}(\alpha),$$

*where $\mathcal{T}(\alpha) \in \mathbb{R}^{m,m}$ is a matrix of scaled critical values corresponding to a given significance level $\alpha$ with zeros along the diagonal, i.e., $\mathcal{T}(\alpha)_{j,j'} = \begin{cases} 0, & \text{if } j = j' \\ \frac{1}{n}\chi^2_{1-\alpha}(1), & \text{otherwise} \end{cases}$, with $\chi^2_{1-\alpha}(1)$ being the $1 - \alpha$ quantile of the chi-square distribution with 1 degree of freedom.*

Notice the similarity between Definitions 2 and 1: if we set $\mathcal{T}(\alpha)$ to be a matrix of 0s and forgo standardization (i.e., use $C$ instead of $Z$), then $\frac{1}{n^2}\sum_{i=1}^{n}\varphi(S_{i,\cdot}) = \hat{\Delta}_{\mathbf{X}}$. Now, the differences: $\hat{\Delta}_{\mathbf{X}}$ is a single matrix computed over an entire set of samples, whereas $\varphi$ is a map that projects each given sample to the new feature space; each entry of $\hat{\Delta}_{\mathbf{X}}$ is simply a distance covariance value, whereas each entry of the sum of $\varphi(S_{i,\cdot})$ over $i$, by using standardization (using $Z$ instead of $C$) and subtracting a critical value, corresponds to the result of using a distance covariance value in a statistical hypothesis test for independence—indeed:

**Lemma 3** *Let $S \in \mathbb{R}^{n,m}$ be a set of $n$ iid samples from random variables $X_1, \ldots, X_m$ with finite first moments. For a given significance level $\alpha$, under the null hypothesis of $X_j \perp\!\!\!\perp X_{j'}$, rejecting $h_\emptyset$ if $\left(\sum_{i=1}^{n}\varphi(S_{i,\cdot})\right)_{j,j'} > 0$ is a statistically consistent test against all types of dependence.*

**Proof** This follows from (Székely and Rizzo, 2009, Theorem 5 and Corollary 2) and how $\varphi$ is defined to correspond to the difference between distance covariance and critical values. ∎

These differences between $\hat{\Delta}_{\mathbf{X}}$ and $\varphi$ serve two important purposes: first, they ensure $\varphi$ maps to a reproducing kernel Hilbert space so that our Definition 9 is a corresponding kernel function (Schölkopf et al., 2001); and second, as the name "dependence contribution map" suggests, they ensure $\varphi(S_{i,\cdot})$ is informative not just about distance covariance but about nonlinear dependence and about how the inclusion of sample $S_{i,\cdot}$ in a set of samples $S$ contributes to the dependence patterns estimated from $S$— this is the key intuition behind how our kernel function is used to learn structurally homogeneous sample subsets, as explicated in the following sections.

Compared to other general measures of dependence, such as mutual information (Shannon, 1948; Cover, 1999) or ball covariance (Pan et al., 2019), we chose the distance covariance in particular because it has mathematical properties that are convenient (if not necessary) for defining the kernel: namely, (1) the pairwise distance correlation matrix can be rewritten in terms of the per-sample covariance (or dependence, if one subtracts a critical value) contribution matrix, which facilitates comparing two samples and thus defining the kernel—this is for example complicated by the logarithms used in defining mutual information; and (2) the distance covariance under the null hypothesis provably approaches the chi-square distribution in the sample limit, facilitating the direct computation of critical values corresponding to a given significance level ($T(\alpha)$ in Definition 2), whereras estimating critical values in the case of mutual information or ball covariance using permutations would be computationally infeasible.

## 2.2. Causal Graphs in Kernel Space

In general, a full causal structure can only be learned with sufficient data about the effects of interventions, and thus causal structure learning from purely observational data is usually possible only up to an equivalence class of causal graphs (Spirtes et al., 2000; Pearl, 2009). For example, the classic PC and IC algorithms, under the assumptions of no selection bias and no confounding by latent variables, do not necessarily return a fully-specified DAG but instead return a mixed graph,

containing possibly directed and undirected edges, representing the Markov equivalence class (Spirtes and Glymour, 1991; Pearl and Verma, 1995).

We now define a set of equivalence classes for ancestral graphs (AGs), which—unlike causal DAGs—do not assume the absence of selection bias and latent confounders (Richardson et al., 2002):

**Definition 4** *Consider an arbitrary ancestral graph $\mathcal{A}$ with the set of vertices $V^{\mathcal{A}}$ and edge function $E^{\mathcal{A}}$, and denote the set of unconditional $m$-connection statements entailed by their corresponding unique maximal ancestral graph as $M^{\mathcal{A}} = \{(j, j') : j \not\perp_m j' \mid \emptyset\} \subseteq V^{\mathcal{A}} \times V^{\mathcal{A}}$. For any ancestral graph $\mathcal{A}'$ such that $V^{\mathcal{A}'} = V^{\mathcal{A}}$, define the unconditional equivalence relation denoted by '$\sim_{\mathrm{U}}$' as*
$$\mathcal{A} \sim_{\mathrm{U}} \mathcal{A}' \quad \text{if and only if} \quad M^{\mathcal{A}} = M^{\mathcal{A}'}.$$

**Lemma 5** *This lemma has two parts: (i) the relation $\sim_{\mathrm{U}}$ is an equivalence relation over the set of ancestral graphs $\mathbb{A}$; (ii) for an arbitrary ancestral graph $\mathcal{A} \in \mathbb{A}$, the bidirected graph $\mathcal{U}^{\mathcal{A}} = (V^{\mathcal{A}}, E^{\mathcal{U}})$, where $E^{\mathcal{U}}$ maps all pairs $(j, j') \in M^{\mathcal{A}}$ to the bidirected edge symbol '$\leftrightarrow$', is a unique representative of the equivalence class $[\mathcal{A}]$.*

**Proof** For (i), recall that an equivalence relation is any relation satisfying reflexivity, symmetry, and transitivity (Devlin, 2003), all of which are satisfied by $\sim_{\mathrm{U}}$ because of its correspondence to the relation '$=$' between sets. Thus, to prove (ii), it suffices to show that the map $s : \mathbb{A}/\sim_{\mathrm{U}} \to \mathbb{A}, [\mathcal{A}] \mapsto \mathcal{U}^{\mathcal{A}}$ is injective (i.e, that it is a *section*) and that $[s([\mathcal{A}])] = [A]$ (Mac Lane, 2013). The key to the proof is the observation that $\mathcal{U}^{\mathcal{A}}$, because it contains only bidirected edges, is maximal and therefore entails exactly the unconditional $m$-separation statements $M^{\mathcal{A}}$, thus by (i) we have $\mathcal{U}^{\mathcal{A}} \sim_{\mathrm{U}} \mathcal{A}$ or equivalently $\mathcal{U}^{\mathcal{A}} \in [\mathcal{A}]$ or equivalently $[\mathcal{U}^{\mathcal{A}}] = [\mathcal{A}]$. Let $\mathcal{A}, \mathcal{A}'$ be arbitrary AGs, and assume $s([\mathcal{A}]) = s([\mathcal{A}'])$. Then by definition of $s$ we have $\mathcal{U}^{\mathcal{A}} = \mathcal{U}^{\mathcal{A}'}$, and by the observation above, $\mathcal{U}^{\mathcal{A}} \in [\mathcal{A}']$ and thus $[\mathcal{A}] = [\mathcal{A}']$, making $s$ injective. And finally, by the definition of $s$ and also by the observation above, $[s([\mathcal{A}])] = [\mathcal{U}^{A}] = [A]$, completing the proof. ∎

This equivalence relation and its representatives has some important but perhaps subtle properties. First, it is different from Markov equivalence over AGs (which is characterized by partial ancestral graphs, PAGs) (Zhang, 2007)—it uses only unconditional $m$-separation while PAGs are learned from conditional $m$-separation statements. Second, because all DAGs are AGs, $\sim_{\mathrm{U}}$ is also an equivalence relation over DAGs. Third, being a representative means that every equivalence class includes exactly one fully bidirected graph (along with other equivalent AGs). Fourth, because each representative is formed by considering $m$-connected paths, $\mathcal{U}^{\mathcal{A}}$ is not equivalent to what would be generated by some "edge-wise" procedure, such as simply replacing every edge in a PAG/AG/DAG/Markov random field/moralized DAG with bidirected edges. Finally, its most important property is that it facilitates Theorem 8, for which we first need a few more definitions.

**Definition 6** *Given arbitrary ancestral graphs $\mathcal{A}, \mathcal{A}' \in \mathbb{A}$ over the same set of vertices, define the Hamming similarity product, denoted '$\bullet$', as $\bullet : \mathbb{A} \times \mathbb{A} \to \mathbb{A}$ and $\mathcal{A} \bullet \mathcal{A}' \mapsto \mathcal{H}$, where $\mathcal{H} = (V^{\mathcal{A}}, E^{\mathcal{H}})$ and the function $E^{\mathcal{H}}(j, j') = '\leftrightarrow'$ if and only if $E^{\mathcal{A}}(j, j') = E^{\mathcal{A}'}(j, j')$.*

In words, the Hamming similarity product between two ancestral graphs returns a fully bidirected graph, with edges only where the two graphs have the same edge type. Now, shifting from ancestral graphs to real-valued square matrices:

**Definition 7** *Let '$\sim_{\mathrm{O}}$' denote the orthant equivalence relation ('orthant' is the generalization of 'quadrant' from $\mathbb{R}^2$ to arbitrarily higher dimensions) in square real matrices, i.e., for matrices $Y, Y' \in \mathbb{R}^{m,m}$ and with the element-wise function* $\mathrm{sign}(Y)_{j,j'} = \begin{cases} 1, & \text{if } Y_{j,j'} > 0 \text{ or } j = j' \\ -1, & \text{otherwise} \end{cases}$,

$Y \sim_{\mathrm{O}} Y'$ *if and only if* $\mathrm{sign}(Y)_{j,j'} = \mathrm{sign}(Y')_{j,j'}$ *for all $j, j'$.*

**Theorem 8** *Let $a$ be the map from the set of unconditional equivalence classes over ancestral graphs with $m$ vertices, $\mathbb{A}^m/\sim_{\mathrm{U}} = \mathbb{U}^m$, to the set of orthant equivalence classes over the image of $\varphi$, i.e., $m \times m$ symmetric real matrices with positive diagonal entries, $\varphi(\mathbb{R}^m)/\sim_{\mathrm{O}} = \mathbb{O}^m$, defined by $a : \mathcal{U} \mapsto O$, where* $O_{j,j'} = \begin{cases} 1, & \text{if } E^{\mathcal{U}}(j,j') = \text{'}\leftrightarrow\text{'} \text{ or } j = j' \\ -1, & \text{otherwise} \end{cases}$ *. Then $a$ is a group isomorphism between $(\mathbb{U}^m, \bullet)$ and $(\mathbb{O}^m, \odot)$, where '$\odot$' denotes the element-wise product.*

**Proof** First, note that $(\mathbb{U}^m, \bullet)$ is indeed a group, satisfying the three group axioms (Artin, 2011): the representative of its identity element is the fully connected bidirected graph over $m$ vertices, $\mathcal{U}^{\mathbb{1}}$; each element is its own inverse; and $\bullet$ is associative. Likewise, $(\mathbb{O}^m, \odot)$ is a group with identity element $[\mathbb{1}^{m,m}]$, each element its own inverse, and the associative element-wise product operator.

Now, to show the two groups are isomorphic, it suffices to show (i) that $a$ is bijective and (ii) that for arbitrary $\mathcal{U}, \mathcal{U}' \in \mathbb{U}^m$, $a(\mathcal{U}) \odot a(\mathcal{U}') = a(\mathcal{U} \bullet \mathcal{U}')$. For (i) notice that if $U \neq U'$, then there must be at least one pair of vertices $j, j'$ such that $E^{\mathcal{U}}(j,j') \neq E^{\mathcal{U}'}(j,j')$ and thus clearly $O_{j,j'} \neq O'_{j,j'}$, so $a$ in injective. Furthermore, notice that every distinct $O \in \mathbb{O}^m$ is the image of some graph $\mathcal{U}$, so $a$ is also surjective. For (ii), for every $j, j' \in \{1, \ldots, m\}$, the definitions of $a$, $\odot$, and $\bullet$ ensure $a(\mathcal{U})_{j,j'} \odot a(\mathcal{U}')_{j,j'} = 1 \iff E^{\mathcal{U}}(j,j') = E^{\mathcal{U}'}(j,j') \iff 1 = a(\mathcal{U} \bullet \mathcal{U}')$, completing the proof. ∎

For causal inference, which (often, but not necessarily) amounts to taking several samples in real space and inferring a single corresponding member in the space of ancestral graphs (or, more often, its quotient set by some equivalence relation), Theorem 8 means we can compare the different graphs of different sample sets without having to first move to the ancestral graph space.

Finally, notice the space of real square matrices is not a typical sample space but rather precisely (a superspace of) the space that our dependence contribution map $\varphi$ (Definition 2) maps samples to—this means that mapping samples with $\varphi$ allows us to make use of the group isomorphism. Though this already provides an intuition for why using $\varphi$ would help with causal clustering, explicitly mapping each sample with it would be unnecessarily computationally expensive, and we are ultimately interested in morphisms between *metric spaces* (not just groups) of samples and graphs. To address this, we therefore now move on to defining a kernel for $\varphi$.

### 2.3. The Dependence Contribution Kernel

**Definition 9** *Let $S, Z, \mathcal{T}$, and $\varphi$ be as in Definition 2. We define the* dependence contribution kernel *using the Frobenius (denoted by the subscript $_{\mathrm{F}}$) inner product and norm:* $\kappa(S_{i,\cdot}, S_{i',\cdot}) = \frac{\langle \varphi(S_{i,\cdot}), \varphi(S_{i',\cdot}) \rangle_{\mathrm{F}}}{\|\varphi(S_{i,\cdot})\|_{\mathrm{F}} \|\varphi(S_{i',\cdot})\|_{\mathrm{F}}}$. *A more convenient expression for applying the kernel to a data set is obtained by first defining a helper kernel, $\gamma$, using it along with* vec *from Definition 1:*

$$\begin{aligned} \gamma(S_{i,\cdot}, S_{i',\cdot}) &= \langle \varphi(S_{i,\cdot}), \varphi(S_{i',\cdot}) \rangle_{\mathrm{F}} \\ &= \left( (\mathrm{vec}(Z_{i,\cdot})^\top \mathrm{vec}(Z_{i',\cdot}))^2 - Z_{i,\cdot} \mathcal{T} Z_{i,\cdot}^\top - Z_{i',\cdot} \mathcal{T} Z_{i',\cdot}^\top + \|\mathcal{T}\|_2^2 \right. \end{aligned}$$

*This allows us to write*

$$\kappa(s, s') = \frac{\gamma(S_{i,\cdot}, S_{i',\cdot})}{\gamma(S_{i,\cdot}, S_{i,\cdot})^{\frac{1}{2}} \gamma(S_{i',\cdot}, S_{i',\cdot})^{\frac{1}{2}}}$$

*Finally, note that $\kappa$ can be readily implemented on an entire set of samples, returning an entire Gram (kernel) matrix instead of a scalar value, by replacing the matrix operations above with tensor operations and specifying the correct axes along which summation occurs—an open source Python implementation can be found at* [https://causal.dev/code/dep_con_kernel.py](https://causal.dev/code/dep_con_kernel.py).

A proper distance metric can also be obtained from this kernel through function composition: $\arccos \circ \kappa$. The key idea behind the kernel is that it is the cosine similarity in the space that $\varphi$ maps to, meaning for arbitrary sample points $x, x'$ it evaluates to $\cos(\theta)$, where $\theta$ is the angle between $\varphi(x)$ and $\varphi(x')$. In this space, $\theta$ represents the dissimilarity of the *dependence patterns* underlying $x$ and $x'$, without being biased by the possibly different magnitudes of $\varphi(x)$ and $\varphi(x')$ due to differing *variances*. Indeed, it can be used as a statistical test of whether samples come from different dependence structures and therefore causal models:

**Theorem 10** *Let $S \in \mathbb{R}^{n,m}$, $S' \in \mathbb{R}^{n',m}$ be sets of $n, n'$ iid samples drawn respectively from the random variables $X = (X_1, \ldots, X_m)$ and $X' = (X'_1, \ldots, X'_m)$ with finite first moments. Then, $\sum_{i=1}^{n} \sum_{i'=1}^{n'} \kappa(S_{i,\cdot}, S'_{i',\cdot}) < 0 \implies \exists j, j' \in \{1, \ldots, m\}$ such that $\mathcal{I}(X_j, \emptyset, X_{j'}) \neq \mathcal{I}(X'_j, \emptyset, X'_{j'})$.*

**Proof** Through Slutsky's Theorem (see Takeshi, 1985, Theorem 3.2.7) and the continuous mapping theorem (see Van der Vaart, 2000, Theorem 2.3), the consistency of $\varphi$ (Lemma 3) guarantees the consistency of $\kappa$. Because the numerator of $\kappa$ is a Frobenius inner product of $\varphi$,

$$\sum_{i=1}^{n} \sum_{i'=1}^{n'} \kappa(S_{i,\cdot}, S'_{i',\cdot}) \propto \sum_{i=1}^{n} \sum_{i'=1}^{n'} \sum_{j=1}^{m} \sum_{j'=1}^{m} \varphi(S_{i,\cdot})_{j,j'} \varphi(S'_{i',\cdot})_{j,j'}.$$

Thus, in order for $\sum_{i,i'} \kappa(S_{i,\cdot}, S'_{i',\cdot}) < 0$, there must be a $j$ and $j'$ for which $\varphi(S_{i,\cdot})_{j,j'} > 0$ but $\varphi(S'_{i',\cdot})_{j,j'} < 0$ (or vice versa), and thus the hypothesis test in Lemma 3 would reject the null hypothesis that $X_j \perp\!\!\!\perp X_{j'}$ but fail to reject that $X'_j \perp\!\!\!\perp X'_{j'}$. ∎

**Corollary 11** *Due to the relationship between independence and causal structure, an immediate result of Theorem 10 is that $\sum_{i,i} \kappa(S_{i,\cdot}, S'_{i',\cdot}) < 0$ implies $X$ and $X'$ have different causal structures.*

**Theorem 12** *Let $d$ be the distance measure between unconditional equivalence classes of ancestral graphs over $m$ vertices, $d(\mathcal{U}, \mathcal{U}') = m^2 - |\{(j, j') : E^{\mathcal{U} \bullet \mathcal{U}'}(j, j') = \text{'}\leftrightarrow\text{'}\}| - m$. For given sample sets $S, S'$ (i.e., real $n \times m$ matrices), use $\bar{\varphi}(S)$ to denote the mean of the sample in kernel space, $\sum_i \varphi(S_{i,\cdot})$, and say $S \sim_{\mathrm{K}} S'$ if and only if $\bar{\varphi}(S) \sim_{\mathrm{O}} \bar{\varphi}(S')$; denote the corresponding quotient set by this equivalence class as $\mathbb{R}^{n,m} / \sim_{\mathrm{K}} = \mathbb{K}^{n,m}$ and a representative from each equivalence class as $Q \in [S]$. Let $\delta$ be the distance between sets of samples in $\mathbb{K}$ defined as $\delta(Q, Q') = m^2 - \frac{1}{2n^2} \sum_{i,i'} \gamma(Q_{i,\cdot}, Q'_{i,\cdot})$. Let $b : \mathbb{U}^m \to \mathbb{K}^{n,m}, b : \mathcal{U} \mapsto \Omega$, where $\Omega$ is the unique element in $\mathbb{K}$ such that $\text{sign}(\bar{\varphi}(\Omega)) = a(\mathcal{U})$. Then $b$ is a distance-preserving map (i.e., an isometry) from the metric space $(\mathbb{U}^m, d)$ to $(\mathbb{K}^{n,m}, \delta)$.*

**Proof**  Notice that $(\mathbb{U}^m, d)$ is indeed a metric space (Choudhary, 1993, Ch. 2): $d(\mathcal{U}, \mathcal{U}') = 0$ iff $\mathcal{U}^{-1} \bullet \mathcal{U}'$ is the empty graph, which happens iff $\mathcal{U} = \mathcal{U}'$; the symmetry of $d$ follows from the symmetry of $\bullet$; and for subadditivity of $d$, observe that for vertices $j, j'$ in arbitrary 2-vertex graphs $\mathcal{U}, \mathcal{U}', \mathcal{U}''$ we have either $d(\mathcal{U}, \mathcal{U}'') = 2$, in which case $d(\mathcal{U}, \mathcal{U}') + d(\mathcal{U}', \mathcal{U}'') = 4$, or we have $d(\mathcal{U}, \mathcal{U}'') = 0$, in which case $d(\mathcal{U}, \mathcal{U}') + d(\mathcal{U}', \mathcal{U}'')$ is either 0 or 4—in both cases $d(\mathcal{U}, \mathcal{U}'') \leq d(\mathcal{U}, \mathcal{U}') + d(\mathcal{U}', \mathcal{U}'')$; this easily extends to graphs of arbitrary numbers of vertices. Likewise, $(\mathbb{K}^{n,m}, \delta)$ is a metric space: $\delta(Q, Q') = 0 \iff \frac{1}{2n^2} \sum_{i,i'} \gamma(Q_{i,\cdot}, Q'_{i,\cdot}) = m^2 \iff \bar{\varphi}(Q)_{j,j'} = \bar{\varphi}(Q)_{j,j'}$, for all $j, j'$, so iff $Q = Q'$; symmetry and subadditivity of $\delta$ follow from the symmetry and subadditivity of $\gamma$.

Finally, to show $b$ is an isometry, we must show (i) that it is bijective and (ii) that for all $\mathcal{U}, \mathcal{U}' \in \mathbf{U}^m$, $d(\mathcal{U}, \mathcal{U}') = \delta(b(\mathcal{U}), b(\mathcal{U}'))$. For (i), observe that by the group isomorphism $a$ and definition of $b$, we have $\mathcal{U} \neq \mathcal{U}' \implies a(\mathcal{U}) \neq a(\mathcal{U}') \implies Q \neq Q' \implies b(\mathcal{U}) \neq b(\mathcal{U}')$ and so $b$ is injective. Also observe that because $\mathbb{K}$ is exactly the set of representatives of orthant equivalence classes of sample sets in kernel space, then for every $Q \in \mathbb{K}$, there exists a $\mathcal{U}$ such that $b(\mathcal{U}) = Q$, and so $b$ is surjective. For (ii), isomorphism $a$ and the relation between element-wise product and Frobenius inner product allow us to write $d(\mathcal{U}, \mathcal{U}') = m^2 - \sum_{j,j'} (O \odot O')_{j,j'} = m^2 - \langle O, O' \rangle_{\mathrm{F}}$. Substituting $O, O'$ with their corresponding $\Omega, \Omega'$, and because the Frobenius inner product is a sesquilinear form, we can write $d(\mathcal{U}, \mathcal{U}') = m^2 - \frac{1}{n^2} \sum_{i,i'} \langle \varphi(\Omega_{i,\cdot}), \varphi(\Omega'_{i,\cdot}) \rangle_{\mathrm{F}}$, which by Definition 9 finally gives us that $d(\mathcal{U}, \mathcal{U}') = \delta(\Omega, \Omega')$, completing the proof.  ∎

In less formal terms, Theorem 12 shows how the space of unconditional equivalence classes of ancestral graphs corresponds to the space of real matrices, which is a common space for samples to lie in. More specifically, it shows how the structure defined by distances between graphs is the same as the structure defined by distances between sets of samples and how this sample distance is related to our kernel $\kappa$. Note that this is much stronger than Theorem 10: not only can $\kappa$ tell us that two sets of samples come from different causal models, it gives a measure of just how different the causal models are, in terms of their differing unconditional nonlinear independencies.

To summarize, we began by defining $\varphi$ (Definition 2), which maps a given data set into a new higher-dimensional feature space. This feature space corresponds to a space of causal graphical models, such that samples which are similar in the new feature space must come from similar causal models (Theorem 8). Our main contribution then is to propose the dependence contribution kernel $\kappa$ (Definition 9). This kernel $\kappa$ is guaranteed not only to tell us that two sets of samples come from different causal models (Theorem 10 and Corollary 11) but furthermore exactly how different the causal models are (Theorem 12), all without the computational expense of explicitly projecting samples or learning causal models. Thus, $\kappa$ is well-suited for addressing the causal clustering problem and ensures that resulting clusters will be structurally homogeneous so that subsequent causal structure learning will be more informative.

## 3. Applications

### 3.1. Synthetic Data

Python code for generating this data and our plots is open source and available at https://causal.dev/code/depcon-kernel-evaluation.tar.gz. One data set consists of 600 samples, 100 each from six random DAGs over 10 variables. For the linear case, we generated random parameters for a structural equation model corresponding to each DAG. We then generated

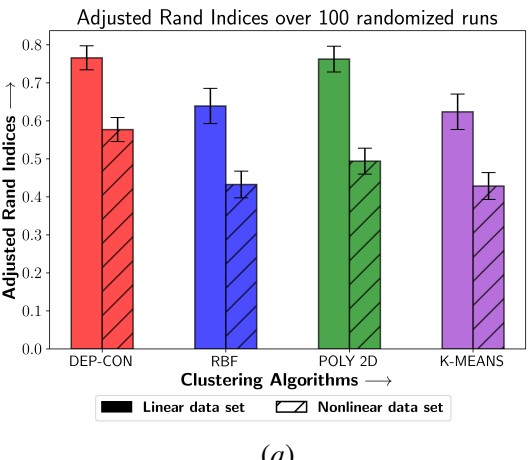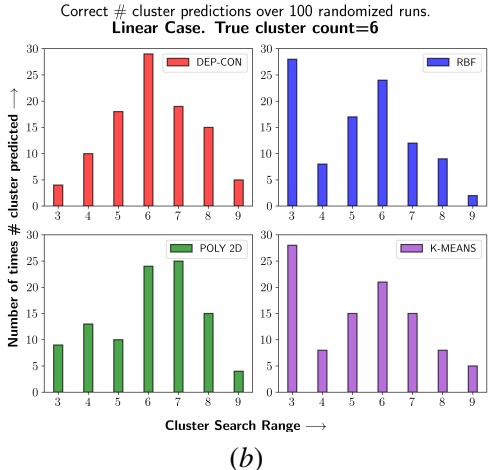

Figure 1: Comparison of clustering with dependence contribution kernel versus baseline methods.

100 of these data sets, and performed clustering on each with kernel $k$-means using our dependence contribution kernel as well as with three baseline methods: (1) the radial basis function (RBF) kernel, for its universality, (2) the $d = 2$ polynomial kernel, for its ability to detect patterns in linear correlation between samples, and (3) plain $k$-means with no kernel, for its simplicity. For the nonlinear case, we then repeated this process, except that we first generated three random DAGs, gave them random parameters for corresponding linear structural equal models (SEMs), and then added a copy of each of the three SEMs but having replaced independencies (so an edge parameter of 0) with nonlinear dependencies.

The linear data set gives a best-case scenario for the the $d = 2$ polynomial kernel baseline method (which looks at the linear correlation patterns between samples), while the nonlinear data set gives a worst-case scenario in which nonlinear dependencies have zero linear correlation and thus are harder to detect with the polynomial kernel baseline method.

For each method on each data set, we chose $k$ for $k$-means by optimizing the Variance Ratio Criterion (VRC) (Caliński and Harabasz, 1974), so that clustering results realistically reflect how the method would perform when the ground truth is not known. We evaluated the clusterings in two ways (Figure 1): first, by using the adjusted Rand index (Rand, 1971) to compare a given clustering to the ground truth, with 1 being a perfect score and 0 being the score by chance (Figure 1(a)); and second, by looking at the predicted number of clusters $k$ found by optimizing the VRC (Figure 1(b)).

The mean adjusted Rand indices show that our kernel performs as well as the best of the baseline methods in the linear case and better than all baseline methods in the nonlinear case (Figure 1(a)). The histograms of the predicted $k$ according to the VRC show that our kernel performs better than all baseline methods in the linear case (Figure 1(b), and the same holds for the unshown nonlinear case).

These results also show that, despite being defined in terms of equivalence classes of causal graphs, our kernel performs reasonably well at clustering samples according to individual DAGs. This is perhaps explained by the fact that the clusters are detectable not only by looking at their causal structure but also by simply looking at their mean in the sample space (indicated by plain $k$-means' performance being better than chance), or perhaps by the fact that different random DAGs tend to belong to different equivalence classes (especially as the number of nodes increases).

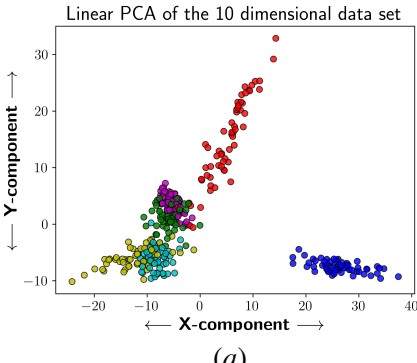
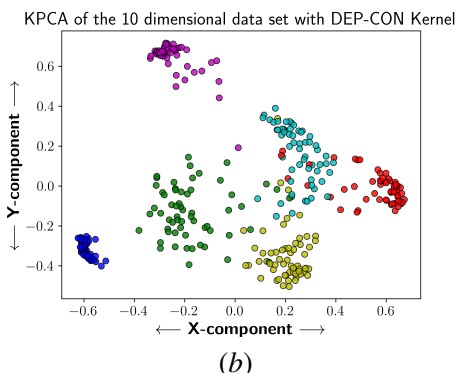

(a)  (b)

Figure 2: Same data visualized with *(a)* linear PCA versus *(b)* dependence contribution kernel PCA.

Furthermore, we can gain some intuition for how the kernel works, i.e., how the samples look in the kernel space and how this representation relates to its clustering performance, by using kernel principal component analysis (kPCA) (Schölkopf et al., 1998). Figure 2(*a*) shows projection onto the first two principal components (according to (linear) PCA) of one of the linear data sets, while Figure 2(*b*) shows the same thing but using kPCA with our kernel. Samples from the different causal models (represented by different colors) are much more easily separated when using our kernel.

### 3.2. Real data

We use kernel $k$-means with our dependence contribution kernel to cluster a gene expression data set and then use the measurement dependence inducing latent (MeDIL) causal model framework for structure learning within each cluster (Markham and Grosse-Wentrup, 2020). The goal of causal clustering here is to reason about the different latent transcription factor (TF) networks governing gene expression (see Verny et al., 2017; Hackett et al., 2020, for other latent causal model approaches to learning TF networks). The original data set comes from Iyer et al. (1999) and can be found at `genome-www.stanford.edu/serum/data/fig2clusterdata.txt`, with subsequent analysis by Dhillon et al. (2003, 2004). Our code is available at `https://causal.dev/code/fibroblast_clustering.py`.

The data consists of the measured gene expression levels of 517 different genes, measured at 11 different time points, i.e., there are 517 samples and 11 different features. In genetics applications, it is not unusual to consider genes to be samples and expression (over time) to be features—indeed the three previous analyses of this data all have this approach—and the intuition is simply that we wish to cluster genes based on patterns in their expression levels over time, in order to identify subsets of genes that are controlled by the same gene regulatory network. Such data exemplifies the structurally heterogeneous populations discussed in Section 1: different genes can of course be regulated by different TFs, and so we can better represent the data by first clustering it into subpopulations that are more homogeneous and then performing causal structure learning on each subpopulation.

For clustering, we used $k = 6$, which we found by looking at both the VRC and the Silhouette Coefficients (Rousseeuw, 1987), computed with the scikit-learn machine learning toolbox (Pedregosa et al., 2011). We implemented (unweighted) kernel $k$-means ourselves, using the pseudocode given

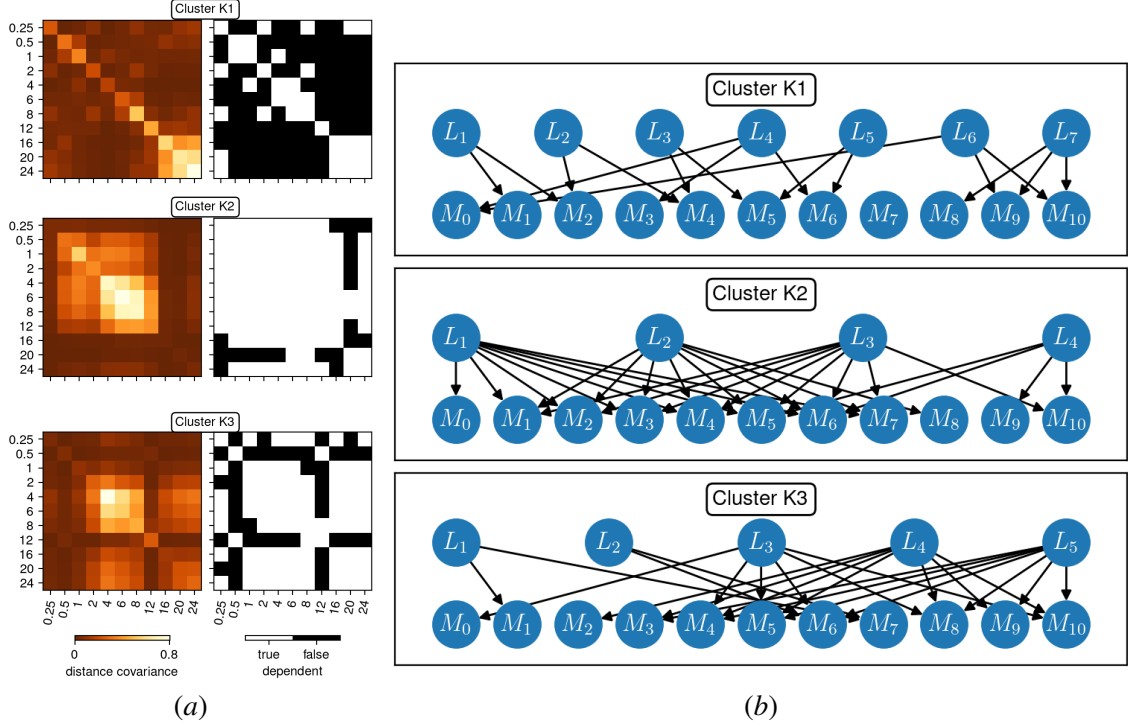

$$(a) \qquad\qquad (b)$$

Figure 3: Results of dependence contribution kernel clustering with significance level $\alpha = 0.1$.

by Dhillon et al. (2004). We then used the MeDIL (Markham et al., 2020) package to learn the dependence structure and latent causal models for each cluster.

Figure 3 shows an example of our results for three of the six gene clusters: Figure 3(a) shows their distance covariance heatmaps and estimated nonlinear dependence structure (so the axes are the 11 different features, i.e., the time, in hours, at which gene expression level was measured), while Figure 3(b) shows their corresponding causal structures, with measurement variables $M_0$–$M_{10}$ for each of the features and learned latent variables $L$ for different posited TFs.

The results show a clear difference in causal structure for the different clusters and allow us to reason about the latent TFs regulating genes in different clusters: notice that the latents in cluster K1 each cause only two or three measurement variables that tend to be close together—e.g., $L_1$ causes $M_1$ and $M_2$, indicating the TF corresponding to $L_1$ is "short-acting", only affecting gene expression from 30 minutes ($M_1$) to 1 hour ($M_2$) after serum exposure; in contrast, the latents in cluster K3 each cause between two and seven measurement variables that tend to be more spread out—e.g., $L_1$ causes $M_1$ and $M_7$, indicating the corresponding TF is more complicated, "long-acting" but not continuously so, affecting gene expression 30 minutes ($M_1$) and 12 hours ($M_7$) after serum exposure, but independently of gene expression in the time between.

Our results are especially noteworthy compared to what happens if one ignores the heterogeneity of the data and learns a single causal structure for the entire data set without first clustering with our kernel into structurally homogeneous subpopulations: in that case, all of the measurement variables are dependent, with a single latent causing them all, and no meaningful conclusions can be drawn about how unmeasured transcription factors regulate measured gene expression, i.e., heterogeneity obscures the underlying causal structures.

In summary, our causal clustering analysis reveals which subpopulations (clusters) of genes have similar latent TF networks as well as how the TF networks differ between clusters—information that is obscured when analyzing the structurally heterogeneous data set as a whole. Additionally, our kernel's ability to measure similarity of nonlinear dependence structure makes it more sensitive than previous analyses of this data set using linear correlation (Dhillon et al., 2003, 2004).

## 4. Discussion

We address the problem of causal clustering—that is, finding the different causal structures underlying a structurally heterogeneous data set. Our main contribution is to develop the *dependence contribution kernel* and prove its suitability for the causal clustering task. This allows us to first use the kernel with existing clustering methods, such as kernel $k$-means or DBSCAN, to identify homogeneous subpopulations. Then we use existing causal structure learning methods on each subpopulation. The kernel guarantees that each subpopulation is more structurally homogeneous and therefore that the learned causal structures better capture the true causal structures within the data than if a single model were learned for the entire heterogeneous population—however, increasingly homogeneous subpopulations comes at the cost of decreasing sample sizes within each subpopulation, so (as is generally the case in clustering) care should be taken when choosing the parameters for whichever clustering method is used.

Furthermore, we prove several interesting theoretical properties of our kernel, including (i) that it can be used as a statistical test for the hypothesis that two sets of samples come from different causal structures, as well as (ii) how it induces a metric space that is isometric to the one defined by Hamming distance between ancestral graphs, i.e., comparing sets of samples with our kernel is equivalent to first estimating the causal graphs of the different sets and then comparing those graphs. Beyond the practical applications of our kernel, as shown by our application in reasoning about latent transcription factor networks that regulate gene expression, this work also draws from and suggests further fruitful connections between a variety of fields, including causal inference, kernel methods, and algebraic statistics. Finally, we emphasize that though clustering is our motivating application in this paper, the dependence contribution kernel can be used in the full variety of machine learning tasks solvable with kernel methods, including, e.g., data visualization/reduction with kernel principal component analysis and classification tasks using kernel support vector machines.

## Acknowledgments

We thank Anja Meunier (University of Vienna) and Liam Solus (KTH Royal Institute of Technology) for helpful discussions and comments on a previous draft. This work is is partially supported by the Wallenberg AI, Autonomous Systems and Software Program (WASP) funded by the Knut and Alice Wallenberg Foundation.

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
