# OpenReview forum: "A Distance Covariance-based Kernel for Nonlinear Causal Clustering in Heterogeneous Populations"
_cclear.cc/CLeaR/2022/Conference — CLeaR 2022 Poster_

### Official Review · Reviewer_trr8 · 2021-11-21

**Confidence:** 3
**Overall Score:** 9

**Main Review:**

Originality: Yes, this paper proposed a novel kernel based method for measuring the similarity between two causal structures. To the best of my knowledge this work is novel and hasn't been solved by existing methods.

Significance: The problem of getting data generated from heterogenous causal structures arises in various settings such as genomics, economics or any other system that can be thought of as comprising various sub modules. And so, the results are of importance to anyone interested in utilizing causal discovery methods, regardless of domain. This paper proposed a new way of looking at the problem of causal clustering using kernels.

Technical quality: Yes, the proposed method seems technically sound. The claims are substantiated by both empirical and theoretical results. To the best of my understanding, the proofs in this paper are correct.
Question for authors: For equivalence classes defined in the traditional causal discovery way, how does this kernel differentiate between two causal graphs within the same equivalence class (maybe I missed this in my reading of the paper)?

Question for authors (I understand this may be very complicated one): Do you have an statistical guarantees for a method to choose the number of causal clusters?

Clarity: Yes - this paper is clear, well written and well explained. The main theme of the paper is conveyed effectively, as well as a intuition behind the method is presented. Great paper overall.

**Summary:**

This paper proposed a novel kernel method to measure similarities between two causal structures. The validity and theoretical properties of this kernel is established, and it's use in a causal clustering setting is demonstrated through a simulated and real world dataset.

---

> ### Author Response · Authors · 2021-12-01
> **The kernel can distinguish between Markov equivalent models by relying on distance covariance values; Statistical guarantees for the number of clusters should be possible to derive**
>
> We thank you for the very positive feedback!
>
> For equivalence classes defined in the traditional causal discovery way (i.e, Markov equivalence), our kernel relies not just on the structural information in the distance covariance matrix (i.e, the binary in/dependence pattern it encodes) but also on the actual covariance values.
> Thus, samples from Markov equivalent models may still be distinguished by our kernel according to the actual distance covariance values.
> In other words, our kernel primarily measures similarity of independence patterns, but it still secondarily measures similarity of distance covariance values.
>
> We do not have any statistical guarantees for choosing the number of causal clusters, however such guarantees should in principle be possible (though difficult) to compute.
> Because the kernel is so explicitly related to statistical hypothesis testing, it should be possible to compute error rates, e.g., on the statement given in Theorem 10.
> For example, consider the element-wise product of two points in the kernel space (note that summing over the resulting matrix would lead to the actual kernel value):
> the probability of the sign (measuring the similarity of the two points in that dependence dimension) being wrong is $2 \alpha(1-\alpha) + 2\beta(1-\beta)$, corresponding to two times (because there are two samples) the probability of a type I error in the test for one sample but not the other plus two times the probability of a type II error for one sample but not the other.
> Next, one must figure out how this error rate for one element of the matrix combines with the other elements (the kernel sums these elements) along with the actual values of the element (the whole matrix is normed), and the result should be the error rate for Theorem 10.
> With these bounds on the error of two samples being incorrectly clustered together, it should be possible to devise a method for picking the number of clusters and finding its corresponding error bounds.

---

### Official Review · Reviewer_LhTe · 2021-11-22

**Confidence:** 3
**Overall Score:** 7

**Main Review:**

reasons to reject
    There are some notion errors in this paper.(e.g section2.1,paragraph2)
    The reason(advantages) for choosing distance covariance could be explained clearly.(There are still some other methods to measure nonlinear dependence)

reasons to accept
    In general, the presented idea seems to be novel and interesting. Identifying heterogeneous causal structures from pooled data is important and could be used in many other applications.
    The motivation of the work is well explained and the pros and cons of different methods are clearly discussed.
    The proof is sufficient.
    The experiment setting is clear and the proposed method performs really good.

**Summary:**

This paper propose the dependence contribution kernel to do causal clustering. DEPCON could identify samples with different latent causal structures in heterogeneous pooled data so that actual learning task could get a more accurate result in each cluster. The proposed method could be used in many problems such as invariant learning.

---

> ### Author Response · Authors · 2021-12-01
> **There are mathematical/convenience reasons for choosing dCov over other nonlinear measures of dependence**
>
> We thank you for the positive feedback!
>
> Thanks also for pointing out that our motivation for choosing distance covariance could be more clearly explained---hopefully the second paragraph of our response to "Reviewer 4ts3" helps to clear it up.

---

### Official Review · Reviewer_4ts3 · 2021-11-24

**Confidence:** 1
**Overall Score:** 6

**Main Review:**

1. An earlier reference for the distance correlation is "Feuerverger, A. (1993), “A consistent test for bivariate dependence,” International Statistical Review/Revue Internationale de Statistique, 61, 419–433."

2. Why do you choose distance covariance in particular, rather than alternative nonparametric metrics of dependence, such as the mutual information, or the ball covariance?

3. Causal structure learning is notoriously different with a large number of covariates. Performing causal structure learning within subgroups identified by the clustering algorithm appears to reduce the sample size even further. There seems to be a tradeoff between homogeneity within each cluster, and the number of observations left for structure learning within each cluster. So my worry is that maybe the best number of clusters for the purpose of clustering, which only takes homogeneity into account, may not yield the best result in terms of structure learning.

As a related comment, in Section 1.1, the authors assume there are enough samples for statistical inference. This does not sound realistic, especially for structure learning.

**Summary:**

The paper proposes to perform clustering before causal structure learning, so as to identify structurally homogenous subsets of samples. To perform such clustering, they propose a distance-covariance based kernel. The main contribution is the proposal of the so-called dependence contribution kernel, so that the resulting kernel space is isometric to the space of causal ancestral graphs.

---

> ### Author Response · Authors · 2021-12-01
> **There are mathematical/convenience reasons for choosing dCov over other nonlinear measures of dependence; Statistical guarantees for the number of clusters should be possible to derive**
>
> Thanks for the earlier reference for distance correlation---we were not aware of that but will add it now to the manuscript.
>
> We chose the distance covariance because it has mathematical properties that are convenient (if not necessary) for defining the kernel:
> namely, (1) the pairwise distance correlation matrix can be rewritten in terms of the per-sample covariance (or dependence, if one subtracts a critical value) contribution matrix, which facilitates comparing two samples and thus defining the kernel---this is for example complicated by the logarithms used in defining mutual information;
> and (2) the distance covariance under the null hypothesis provably approaches the chi-square distribution in the sample limit, facilitating the direct computation of critical values corresponding to a given significance level (T(\alpha) in Definition 2), whereras estimating critical values in the case of mutual information or ball covariance using permutations would be computationally infeasible.
> However, we are admittedly more familiar with distance covariance than other measures, so perhaps defining an analogous kernel for other measures is more straightforward than we realize.
> Our hope is, having defined such a kernel using distance covariance, it is now easier for us and other researchers to similarly define kernels based on other measures of dependence.
>
> Thanks for pointing out the difficult "tradeoff between homogeneity within each cluster, and the number of observations left for structure learning within each cluster".
> This is an important problem not just in our setting here, but in clustering generally.
> In cases where the data is known to be heterogeneous, the best that can be done is to find the correct number of clusters and assignment of samples to the clusters, and our method offers an effective way of doing this (for which statistical guarantees should even be possible to derive---see the last paragraph of our response to "Reviewer trr8")

---

### Decision · Program_Chairs · 2022-01-12

**Decision:**

Accept (Poster)

**Comment:**

This paper proposes to use a distance covariance-based kernel to measure the similarity between two nonlinear independence structures for nonlinear clustering. They prove that the corresponding feature map is a statistically consistent estimation of nonlinear independence structure. They further showed that the kernel space is isometric to the causal ancestral graphs. The reviewers appreciate these theoretical results.

Their method classifies observations based on their nonlinear independence structures, and the nonlinear independence structures correspond to unconditional equivalence classes over ancestral graphs. Still, I wonder if it does not necessarily mean they classify observations on their causal structures. Each equivalence class will include different causal structures. If this is so, the title, causal clustering, might sound exaggerated. I want the authors to clarify this point. I also want the authors to clearly state the limitations raised by the reviewers.